# The Effect of Rolling Texture on the Plastic Deformation of Nano-Gradient Aluminum

**DOI:** 10.3390/nano13152214

**Published:** 2023-07-30

**Authors:** Yaxin Zhang, Hao Lyu

**Affiliations:** College of Transportation Engineering, Dalian Maritime University, Dalian 116026, China; z2095@dlmu.edu.cn

**Keywords:** dislocations, gradient aluminum, texture

## Abstract

Creating alloys with a gradient microstructure in grain size has been shown to be a potential method to resolve the trade-off dilemma between strength and ductility. However, different textures developed with various processing methods cannot be fully eliminated, which can significantly affect the mechanical behavior of alloys. In this study, we use a multiscale framework based on dislocation theory to investigate how the combination of rolling texture and gradient in grain size affects the plastic deformation of nano-gradient aluminum during a tensile test. We found that specific rolling textures, such as {110} texture, can significantly enhance the strength and ductility of nano-gradient aluminum. This improvement is the result of the grain being reoriented and the redistribution of stress and strain, which are caused by the combined influence of texture and variation in grain size. These results provide new insights into developing high-performance aluminum by mediating texture and grain size gradient.

## 1. Introduction

Gradient nanomaterials have attracted considerable attention because of their high strength and good ductility [1,2]. From the strength-ductility curve in [3], it is clear that the gradient materials are balanced due to their mutual strength-ductility exclusivity. In gradient materials, their grain size progressively increases from the surface to the interior [4]. This grain size gradient can be obtained by many processing methods, such as surface mechanical attrition treatment (SMAT) [5,6] and surface mechanical grinding treatment (SMGT) [7,8,9], ultrasonic shot peening (USP) [10,11], accumulative roll-bonding (ARB) [12], and high-pressure torsion (HPT) [13,14]. However, not only is this gradient structure expressed in grain size, but the crystallographic texture also constitutes a texture gradient that has an impact on the ductility of the material.

The excellent properties of gradient nanomaterials are closely related to grain effects. Lu et al. [4] found that gradient copper has high strength while maintaining good plasticity, with mechanical forces driving the growth of grains in the gradient layers. Zhu et al. [15] found that the gradient structure produces unique additional strain hardening, which leads to high plasticity, and that grain size gradients in tension produce macroscopic strain gradients along the depth of the gradient due to incompatibility, converting uniaxial stresses into multi-axial stresses. Cheng et al. [16] suggest that plastic strain begins in the softest region where the initial yield strength is lowest and extends to the harder region with increasing load. However, less effort has been put into studying the effect of texture that is unavoidably created by specific processing methods.

Texture changes are influenced by the processing method and have a certain pattern [17,18]. Recent experiments have demonstrated that the processing-induced texture gradient can have a significant effect on the mechanical behavior of alloys. Chen et al. [19] found the existence of a typical strong β-fiber texture in heavy extruded pure aluminum with a heterogeneous structure; moreover, they described the changes of various texture contents during the annealing process. Subsequently, Chen et al. [20] revealed that texture varies from the surface to the center in heterogeneously structured samples, indicating the presence of textural heterogeneity. SMAT-treated Mg can also produce texture structure gradients [21]. Zhu et al. [22] also reported the appearance of texture gradient in SMAT-treated mild steel that can affect plastic deformation. Kuang et al. [23] declared that the texture gradient without a large grain size gradient in Al-Mn strips by double-roll casting can effectively improve the material ductility without sacrificing strength. Thus, multiple complex gradient microstructures containing grain size gradients and texture gradients [24,25] are more likely to achieve superior strength-ductility properties than single-gradient materials.

To study a multiscale microstructure effectively, an efficient numerical simulation that takes into account the grain size effect and a large number of grains is necessary. On the one hand, size effects can usually be captured by introducing the intrinsic length scale within geometrically necessary dislocations (GND) [26,27], which must be present when plastic strain gradients are present in the material during non-uniform straining. On the other hand, the stress gradient aroused by dislocation pile-ups against grain boundaries under nonhomogeneous shearing can also significantly affect the stress-strain response [28]. Recently, Liu et al. [29] developed a rate-dependent crystal plasticity finite element model by accounting for the grain size effect and cohesive zone model to investigate the plastic deformation and crystal crack of a gradient aluminum. Lu et al. [30] adopt a crystal plasticity model to study the deformation of gradient nanostructured TWIP steel. Zhang et al. [31] also developed a nonlocal crystal plasticity model to study gradient grained materials by accounting for geometrically necessary dislocations and related kinematic hardening, which can simulate more grains. Similarly, the grain size effect has also been implemented in the mean field crystal plasticity model [32,33]. One example is the viscoplastic self-consistent model (VPSC) developed by Lebenson and Tome [34], which can quickly simulate a large number of grains.

Therefore, the purpose of this study is to explore how the initial texture of rolling fibers, which share a common rolled texture, affects the plastic deformation of gradient aluminum. To accomplish this, we will utilize a multiscale framework that has been previously developed based on dislocation activities [35]. In order to compare the plastic deformation and overall mechanical behavior between samples with randomly oriented grains and those with rolling textures, tensile tests were performed on specimens containing different volume fractions of various texture fiber components. The effects of gradient depth and the content of pronounced texture fibers on the plastic deformation mechanisms of gradient aluminum were analyzed in detail and discussed.

## 2. Materials and Methods

### 2.1. VPSC-CDD Model

The pre-proposed multiscale framework [35] couples a dislocation density evolution law with the VPSC model [36]. The VPSC model enables the prediction of macroscopic mechanical behavior and texture evolution for polycrystalline aggregates represented by grains with orientation and volume fractions. This modeling approach considers each constituent grain as an inclusion embedded within a homogeneous viscoplastic medium [36]. The VPSC is able to capture the mechanical behavior and texture evolution of FCC crystals [37,38,39]. The detailed continuum mechanics and the homogenization method of the VPSC can be referred to in Appendix A. Here, the Orowan relation is used to bridge up the evolution of dislocation and local deformation by evaluating the motion of the mobile dislocations as follows:(1)$${\dot{\gamma }}^{s}=\rho_{M}^{s}b{\bar{v}}_{g}^{s}$$

The plastic shear rate ${\dot{\gamma }}^{s}$ in the slip system *s* is originally expressed as a power law of stress in the VPSC model, $\rho_{M}^{s}$ is the density of moving dislocations in the slip system s, b is the Burgers vector, and ${\bar{v}}_{g}^{s}$ is the average dislocation glide velocity at each time step within each grain and can be expressed as:(2)$${\bar{v}}_{g}^{s}=\left\{\begin{array}{c} v_{0}{\left|\frac{\tau^{s}}{\tau_{CR}^{s}}\right|}^{\frac{1}{r}}sign(\tau^{s})\tau^{s}\geq \tau_{CR}^{s}\\ 0\;\tau^{s}\leq \tau_{CR}^{s} \end{array}\right.$$
where the reference velocity $v_{0}$ is a constant in the order of $10^{-4}$, r is the strain rate sensitivity, $\tau^{s}$ is the resolved shear stress on the slip system *s*, and $\tau_{CR}^{s}$ stands for the critical resolved shear stress on the slip system. When $\tau^{s}$ is larger than $\tau_{CR}^{s}$, slip is activated, and plastic deformation can occur. According to Schmidt’s law, $\tau^{s}$ is obtained by decomposing the local stress into the corresponding slip system *s*, and $\tau_{CR}^{s}$ can be decomposed as follows:(3)$$\tau_{CR}^{s}=\tau_{0}^{s}+\tau_{H}^{s}+\tau_{SG}^{s}$$
where $\tau_{0}^{s}$ represents the friction of lattice, while $\tau_{H}^{s}$ denotes a hardening term due to interactions between dislocations, and $\tau_{SG}^{s}$ is a size-dependent term. Here, instead of using the empirical hardening law in the original VPSC, the hardening caused due to dislocation activities can be described using these three terms. The hardening caused by dislocation interactions is described by Bailey–Hirsch hardening [40]. Thus, $\tau_{H}^{s}$ can be expressed as:(4)$$\tau_{H}^{s}=\sum_{\beta =1}^{N}\Omega^{s\beta }c*\mu b\sqrt{\rho_{T}^{\beta }}$$
where c* is a constant in the order of 1, μ is the shear modulus, and $\rho_{T}^{\beta }$ is the total statistically stored dislocation (SSD) density on the slip system β. The relationship between slip s and β strength can be found using the interaction matrix. This matrix can be obtained by performing discrete dislocation dynamics (DDD) simulations. The total statistically stored dislocation density $\rho_{T}^{\beta }$ can be divided into two components: mobile dislocation density $\rho_{M}^{\beta }$ and immobile dislocation density $\rho_{IM}^{\beta }$. Based on different mechanisms of dislocation activities, both mobile and immobile dislocation density evolve with time and can be expressed as:(5)$${\dot{\rho }}_{M}^{s}=q_{1}\rho_{M}^{s}\frac{{\bar{v}}_{g}^{s}}{{\tilde{l}}_{g}^{s}}-q_{2}2R_{c}{\left(\rho_{M}^{s}\right)}^{2}{\bar{v}}_{g}^{s}-q_{3}\rho_{M}^{s}\frac{{\bar{v}}_{g}^{s}}{{\tilde{l}}_{g}^{s}}+q_{4}{\left(\frac{\tau^{s}}{\tau_{CR}^{s}}\right)}^{1/r}\rho_{IM}^{s}\frac{{\bar{v}}_{g}^{s}}{{\tilde{l}}_{g}^{s}}+\;q_{5}\sum_{\beta =1}^{N}P^{s\beta }\rho_{M}^{s}\frac{{\bar{v}}_{g}^{s}}{{\tilde{l}}_{g}^{s}}-q_{6}R_{c}\rho_{M}^{s}\rho_{IM}^{s}{\bar{v}}_{g}^{s}-q_{7}R_{c}^{3}\rho_{M}^{s}{\left(\rho_{IM}^{s}\right)}^{2}{\bar{v}}_{g}^{s}\pm p{\bar{v}}_{g}^{s}\cdot \nabla \rho_{M}^{s}$$
(6)$${\dot{\rho }}_{IM}^{s}=q_{3}\rho_{M}^{s}\frac{{\bar{v}}_{g}^{s}}{{\tilde{l}}_{g}^{s}}-q_{4}{\left(\frac{\tau^{s}}{\tau_{CR}^{s}}\right)}^{1/r}\rho_{IM}^{s}\frac{{\bar{v}}_{g}^{s}}{{\tilde{l}}_{g}^{s}}-q_{6}R_{c}\rho_{M}^{s}\rho_{IM}^{s}{\bar{v}}_{g}^{s}+q_{7}R_{c}^{3}\rho_{M}^{s}{\left(\rho_{IM}^{s}\right)}^{2}{\bar{v}}_{g}^{s}$$
where $q_{1}$–$q_{7}$ are the coefficients of seven different terms representing different dislocation interaction mechanisms. The values of these coefficients can be determined by fitting DDD simulations or experiments [41]. The validation of the chosen parameters can be referred to [35]. $R_{c}$ is the critical radius for dislocation interaction. $P^{s\beta }$ is an *N* by *N* matrix that represents the probability of cross-slipping from slip system s to β, in f.c.c *N* = 2. p is the probability of the activation of slip transmission. ${\tilde{l}}_{g}^{s}$ is the mean free path of the sliding dislocation. Here, the first term refers to the multiplication and accumulation of mobile dislocations, such as a Frank-Read source. The second term describes the annihilation of two mobile dislocations with opposite signs. The third term denotes the immobilization of mobile dislocations due to the formation of jogs and junctions. The fourth term relates to the mobilization of immobile dislocations. The fifth term accounts for the cross-slip of screw dislocations, which can be determined using Monte Carlo simulations. The sixth term represents the absorption of mobile dislocations or the emission of immobile dislocations by grain boundaries. We also consider the formation of dislocation dipoles at a rate of $q_{7}R_{c}^{3}\rho_{M}^{s}{\left(\rho_{IM}^{s}\right)}^{2}{\bar{v}}_{g}^{s}$. In addition, the dislocation flux term $\pm p{\bar{v}}_{g}^{s}\cdot \nabla \rho_{M}^{s}$ is also considered, the implementation of which can be found in Appendix B.

The strength of the materials is dependent on the density of dislocations. To introduce the grain size effect, both stress/strain gradient models are considered. On the one hand, both the total statistically stored dislocations $\rho_{T}^{\beta }$ and geometrically necessary dislocations (GNDs) contribute to hardening by resisting the motion of dislocations. GNDs are considered obstacles existing in the grain boundaries when the grain undertakes non-unform strain. In this work, the resistance of GNDs on mobile dislocations is accounted for in the term ${\tilde{l}}_{g}^{s}$, which is expressed as follows:(7)$${\tilde{l}}_{g}^{s}=\frac{c}{\sqrt{\sum w^{\beta s}(\rho_{T}^{\beta }+||\rho_{GND}^{\beta }||)}}$$
where c is a numerical factor in the order of 1 and $w^{\beta s}$ is a matrix similar to the interaction matrix Ω. Here the value of $w^{\beta s}$ is all set as 1. The norm of the GNDs density $\left||\rho_{GND}^{\beta }|\right|$ can be determined by:(8)$$\left|\left|\rho_{GND}\right|\right|=\frac{1}{b}\sqrt{AA}$$
where A represents the Nyes tensor [42], which can be approximated by:(9)$$A=curl(-F^{P})$$

The intrinsic length scale parameter was included in the curl operator of the plastic distortion rate tensor $F^{P}$ predicted by the VPSC model.

On the other hand, stress gradient plasticity models describe the dependence of flow stress on obstacle spacing and a high-order stress gradient caused by dislocation pile-ups against grain boundaries under inhomogeneous shear stress [43]. Here, a simplified linear stress gradient model is employed, and then the size-dependent term can be expressed as:(10)$$\tau_{SG}=\frac{K}{\sqrt{L}}(1+\frac{L^{\prime }}{4\bar{\tau }}|\nabla \bar{\tau }|)$$
where K is the Hall-Petch constant, L is the grain size, $L^{\prime }$ represents the average distance between obstacles (grain boundaries), $\bar{\tau }$ is the equivalent stress, and the stress gradient term $\nabla \bar{\tau }$ introduces the intrinsic length scale. We assume $L^{\prime }$ is identical to L; in this case, the gradient term $\frac{L^{\prime }}{4\bar{\tau }}|\nabla \bar{\tau }|$ becomes a hardening term to the Hall-Petch relation. It depends on the local stress state and its variation across neighboring grains due to crystallographic orientation and dislocation density. When the grain is subjected to homogeneous stress (constant stress), this term vanishes, and the size-dependent term becomes the Hall-Petch relation. Further details can be found in [43].

### 2.2. Implementation

Since the viscoplastic self-consistent (VPSC) model is dimensionless and the approximation of stress/strain gradients requires additional microstructural information, a 2D Voronoi tessellation was utilized to represent the polycrystalline sample. Each Voronoi cell represents an individual grain with corresponding data from the VPSC calculations, including the local stress/strain state, dislocation density evolution, and statistical attributes, such as grain size, neighbor configurations, and crystallographic texture. The spatial gradients of stress and strain could then be evaluated via a moving least squares interpolation method using the stress and strain data from grains within a defined region. Further details of the gradient approximation approach can be found in Reference [44].

### 2.3. Materials

In the present study, the influence of the initial rolling texture on the strength and ductility of gradient aluminum was examined using a multiscale modeling framework incorporating dislocation-based plasticity [35]. A polycrystalline sample with a grain size gradient and a dimension of 100 μm × 100 μm was fabricated using Voronoi tessellation [44]. The sample was then divided into three regions based on grain size along the y-direction: the surface nanocrystalline region (NG) with ultrafine grains (~200 nm), the transition region (TG) with a size gradient connecting the surface nanocrystalline region to the central coarse crystalline region, and the coarse crystalline region (CG) with a grain size of ~10 μm. In addition, by dividing the sample into two surface areas within *D* and a central area, rolling texture is assigned to the grains in the area within *D* (see Figure 1a). A noise of [−5°, 5°] was accounted for in the Euler angles of specific rolling texture.

In the first investigation, the effect of thickness *D* was studied by fixing the specific rolling texture (as shown in Table 1) and varying *D* from 10 μm to 40 μm. In the second study, the different rolling textures of the FCC were assigned at *D* = 40 μm, with the grain orientation in the central region set to a random texture. An additional sample with all grains having a random grain orientation was set as the reference case. The pole figures of the different rolling textures of the FCC are shown in Figure 1.

## 3. Results

In this work, tensile tests were carried out on gradient aluminum samples with different initial texture gradients along the y-direction compared to a reference case with randomly assigned grain orientations. The strain rate was set as $10^{-3}$ s^−1^ and all the parameters are listed in Table 2, which has already been validated in our previous work [35].

In the first investigation, the depth *D* was varied from 10 μm to 40 μm for different rolling textures. Each sample was annotated with a number from 1 to 4, added to the texture symbol, to indicate its depth, i.e., sample *C*1 with texture C at a depth of 10 μm. As shown in Figure 2a, a larger area of texture {C} results in a lower yield strength and tensile strength, while samples with other textures demonstrate an opposite trend, showing an increase of yield strength and tensile strength with increasing *D*. In Figure 2b, sample B3 has a similar strain hardening trend in the initial strain stage as that of the reference case but exhibits a delayed instability (approximately $\frac{d\sigma_{T}}{d\varepsilon_{T}}=\sigma_{T}$) at 0.3. Both samples B3 and B4 denote a better combination of strength and ductility than the reference case. In Figure 2c, one can say that texture {S} initially exhibits much weaker strain hardening than the reference case, but with increasing *D* to 40 μm, sample S4 demonstrates a relatively higher tensile strength and better ductility than the reference case. Figure 2d shows that increasing *D* could result in an earlier occurrence of instability than the reference case.

In the second investigation, the area of the strong rolling texture was set within a depth of 40 μm. The stress-strain responses of samples with different textures are shown in Figure 3. It is obvious that the initial texture {B} can significantly enhance the mechanical behavior of the gradient sample, which is 1.3 times the strength of the one with texture {C}. The texture {S} can somehow affect the mechanical behavior but not dissolve the strength-ductility trade-off dilemma. The existence of texture {C} and {M} can result in lower strength and earlier instability than the reference case.

When comparing the random initial texture to the pole figures at 10% strain, it is apparent that the texture undergoes significant changes in both direction and strength. Specifically, the texture strength increases by more than a factor of 7. For more details, see Appendix C Figure A1. As shown in Figure 4, we can observe a remarkable change in the grain orientation of the texture {C} from the original state to the loading direction, compared to the textures {B} and {S} and {M} where a small number of the grains are not reoriented to the loading direction. Furthermore, it was observed that the intensity of loading direction alignment is approximately 1.2 times higher in texture {C} than in the other three configurations. This implies that more grains are reoriented to the loading direction, resulting in a rapid increase in the slip. Although the profiles are very similar between the textured {C} and random texture structures at 10% strain, the intensity in the [010] direction is still significantly higher in the former than in the latter.

To obtain the average equivalent strain, stress, and mobile dislocation density, the strain, stress, and dislocation density values from each grain are interpolated onto a uniform grid (2000 by 2000) of query points using natural neighbor interpolation. This creates a continuous field of values across the grid. The interpolated value at each query point is then averaged along the X-direction. This results in a 1D profile showing the average value as a function of position along X. The equivalent stress and strain are approximated using the Von Mises yield criterion. Specifically, the Von Mises equivalent stress $\sigma_{eq}$ and equivalent strain $\varepsilon_{eq}$ are calculated as:(11)$$\sigma_{eq}=\sqrt{\frac{1}{2}\left[{\left(\sigma_{11}-\sigma_{22}\right)}^{2}+{\left(\sigma_{22}-\sigma_{33}\right)}^{2}+{\left(\sigma_{33}-\sigma_{11}\right)}^{2}\right]+3(\sigma_{12}^{2}+\sigma_{23}^{2}+\sigma_{31}^{2})}$$
(12)$$\varepsilon_{eq}=\frac{\sqrt{2}}{3}\sqrt{\left[{\left(\varepsilon_{11}-\varepsilon_{22}\right)}^{2}+{\left(\varepsilon_{22}-\varepsilon_{33}\right)}^{2}+{\left(\varepsilon_{33}-\varepsilon_{11}\right)}^{2}\right]+3(\varepsilon_{12}^{2}+\varepsilon_{23}^{2}+\varepsilon_{31}^{2})}$$
where $\sigma_{ij}$, $\varepsilon_{ij}$ represent the components of stress tensor and strain tensor, respectively. In future work, the Hosford yield criterion will be considered an alternate equivalent stress measure.

Figure 5 shows the average equivalent strain/stress and mobile dislocation density along the X-direction versus the grain size gradient direction Y for the 10%, 20%, and 30% strain stages at *D* = 40 μm. In all textures, the grains in the NG region are very small, and the dislocations are difficult to slip, which makes it harder for the grains to deform plastically than those in the CG region. Central grains undergo more plastic deformation compared to other grains, indicating a higher strain gradient from the surface to the center, particularly in the case with texture {B}. All samples display lower strain and dislocation density, but a high stress state near the surface. Samples with texture {B} exhibit a relatively slow strain gradient, followed by a steep strain gradient when Y is around 30~40 μm. This suggests that in the nanoscale, grains with a strong rolling texture {B} may experience less plastic deformation than other cases (refer to the strain and mobile dislocation density distribution in Figure 5). However, at the borders of the strong texture region and random texture region, grains undergo more plastic deformation. This also results in a clear change in the stress gradient at Y = 30~40 μm with further straining to 10% and 20%. In contrast, samples with texture {M} indicate a higher strain gradient along Y initially, followed by a slower strain gradient when Y is around 30~40 μm. It is also worth noting that the stress gradient vs. Y of samples with texture {M} is identical to that of samples with texture {B}. The trend in the case of texture {S} is quite similar to that of the reference case. Samples with texture {C} exhibit a comparable strain distribution along Y, but significantly lower stress levels when compared to the other cases. These trends also hold for the group of samples with strong texture within *D* = 30 μm (Figure 6), but not for the cases with *D* = 10 μm and 20 μm (see Appendix C Figure A2 and Figure A3). When *D* is as thick as the thickness of the nano-grains region close to the surface, the effect of the initial texture is eliminated.

## 4. Discussion

Experimental studies have shown that different textures in gradient materials have different effects on strength and ductility. In particular, according to Kuang et al. [23] the creation of gradient texture in Al-Mn strips can impact local back stress and slow down the fracture process in order to enhance the ductility of the alloy. Wang et al. [48] suggested that heterogeneous deformation-induced hardening due to the dislocation gradient structure significantly increased the strength with little reduction in ductility. Moering et al. [49] also noted that out-of-plane {111} line texture may result in different initial stress states and strain hardening of the material, thereby enhancing the mechanical properties of gradient aluminum rods. Additionally, they noticed [22] the presence of shear texture gradients in SMATed mild steel, which enhances its strength and ductility. According to these studies, the way in which the size of the grains in a material changes along with changes in its texture can have a synergetic effect on its strength and ductility. Though research has identified specific textures and texture gradients that impact material properties, less work has been done on investigating the effect of specific texture fibers in the plastic deformation of gradient aluminum. The present study exhibits that a {011} initial crystallographic texture ({B}) substantially increased the strength of the gradient aluminum and postponed the onset of plastic instability. However, the strengthening effect of the {011} texture is apparent when the oriented grains comprise a substantial volume fraction within the gradient specimen. The Schmid factors of all slip systems for NG and TG grains with different textures are shown in Table 3. The Schmid factor of {001} texture is significantly higher than the other groups, indicating that the texture {C} is most likely to initiate slip in NG and TG regions. On the other hand, in cases with texture {011}, NG and TG grains can barely deform plastically, resulting in a lower equivalent strain close to the surface, as shown in Figure 6. As strain occurs, the texture interacts with the random initial texture to rotate the grains to a relatively stable position [16]. For grains with such a texture, grain rotation may not occur, as there is a high likelihood that the loading direction is directly in the slip direction. After further straining to 10%, polar figures still show a similar trend to the initial one, but with much higher intensity. The Schmid factor in the case of {123} texture ({S}) is slightly higher than that of the {112} texture ({M}). Interestingly, at 10% strain, the pole figures of these two textures and texture {B} are almost identical, as shown in Figure 4b–d. Moreover, it is evident that texture B has a much smaller intensity, indicating that fewer grains have been reoriented in that direction due to less activation of slip in NG and TG grains. Table 4 shows the Schmidt factors of all slip systems for NG and TG grains with different textures at 10% strain. The Schmidt factors of all slip systems for {011} texture have significantly increased, and the summation of the absolute value is the largest among all textures. Therefore, this texture would lead to a higher resolved shear stress (RSS) for the activation of slips. Since a large number of grains with this orientation are located either in NG or TG, the activation of slips depends solely on the grain size effect. When slip is activated in TG grains near the center, the strain gradient at Y = 40 μm decreases, as shown in Figure 6, for 10% and 20% straining. At the same time, the increased dislocation density results in a relatively small stress gradient at the same Y position. So basically, a texture like {011} can lead to a stress/strain redistribution which results in much higher tensile strength and ductility. The combined effect of the starting texture and the grain size results in NG grains that are extremely hard and resistant to plastic deformation, causing a significant strain and stress gradient between the surface and the center. The evolution of the texture can also mediate deformation, leading to stress/strain redistribution. The location of the apparent change in the stress-strain gradient coincides with the border of the NG and TG and the border of grains with strong texture and random texture. This sheds new light on developing gradient aluminum with superior strength and ductility by manipulating both texture and grain size.

In comparison to the experimental results of Chen et al. [19], the current modeling work revealed that an optimal balance of strength and ductility could be attained by tuning the volume fraction of the {011}<112> oriented grains. A recent investigation by Kuang et al. [23] reported that the Al-Mn strip with a specific texture gradient, such as {011}<112>, but no grain size gradient, can possess higher ductility without sacrificing its strength. In addition, their findings [20] provide strong evidence that the {100}<001> texture resulting from extended annealing considerably reduces the strength of aluminum. These results are in line with the outcomes of our simulations.

Furthermore, it should be emphasized that varying the thicknesses of the regions with a specific texture can have a significant impact on the plastic deformation of the gradient aluminum. When *D* approaches the thickness of the NG region, the effect of the initial texture is eliminated. Further studies on mediating the texture gradient and grain size gradient can lead to possible advances in the strength and ductility of gradient materials.

## 5. Conclusions

This study uses a dislocation-based multiscale framework to examine the synergetic effect of the initial rolling texture and grain size on the plastic deformation of gradient aluminum. Grains located at the surface region of gradient aluminum samples are assigned different initial rolling textures. It was found that this initial rolling texture can have a significant impact on the macroscopic behavior of the gradient aluminum. Noteworthy results include:(i)The thickness of the surface region with a specific rolling texture can affect the macroscopic behavior of the gradient aluminum. Once the thickness reaches the thickness of the NG region, the effect of the initial texture is eliminated.(ii)Among all rolling textures, it is found that the {011} (B) texture can lead to superior strength and ductility by causing large stress/strain gradients and stress/strain redistribution due to grain reorientation. Conversely, the {010} (C) texture can result in reduced strength and ductility. Our results are in good agreement with the experimental observations.(iii)This study demonstrates that mediating texture and grain size gradient can lead to further improvements in the strength and ductility of gradient aluminum.

## Figures and Tables

**Figure 1 nanomaterials-13-02214-f001:**
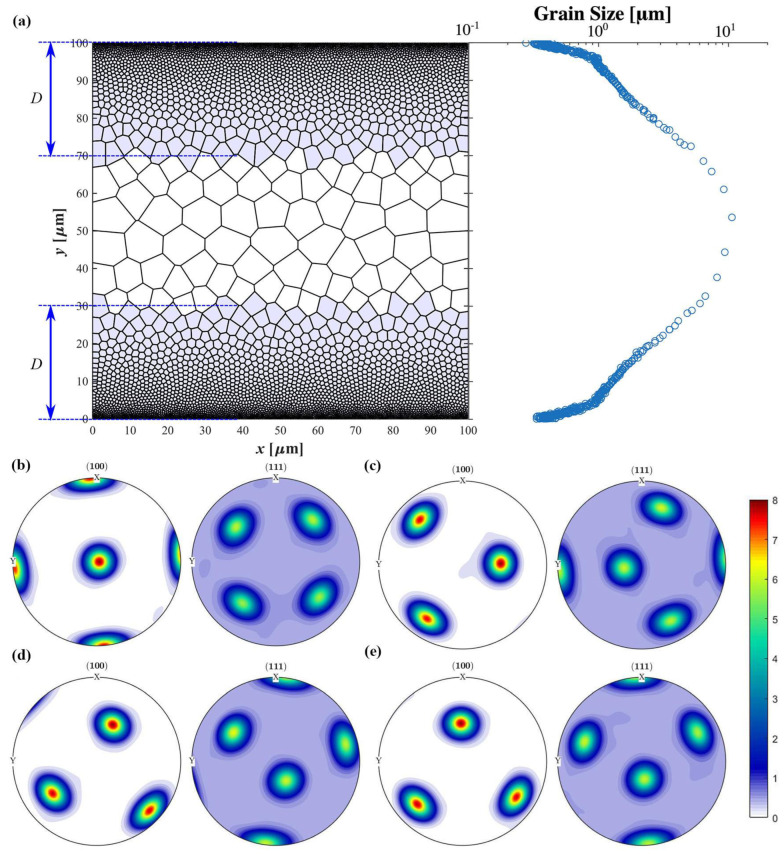
(**a**) Gradient aluminum sample with assigned rolling texture (blue shading area with a thickness of *D* = 30 μm). Pole figures of various initial rolling textures: (**b**) {C}; (**c**) {B}; (**d**) {S}; and (**e**) {M}.

**Figure 2 nanomaterials-13-02214-f002:**
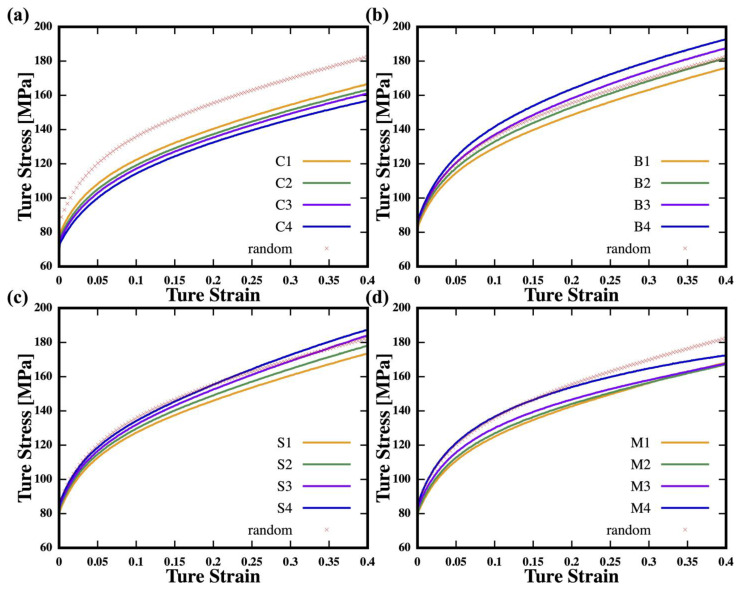
Stress–strain curves for various values of *D* with texture (**a**) {C}, (**b**) {B}, (**c**) {S}, and (**d**) {M}.

**Figure 3 nanomaterials-13-02214-f003:**
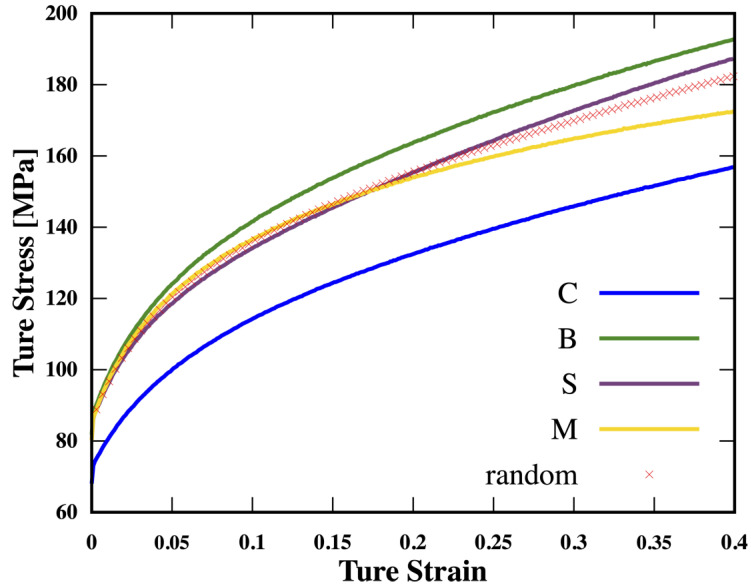
Stress–strain curves for different rolling textures at *D* = 40 μm and the initial random texture.

**Figure 4 nanomaterials-13-02214-f004:**
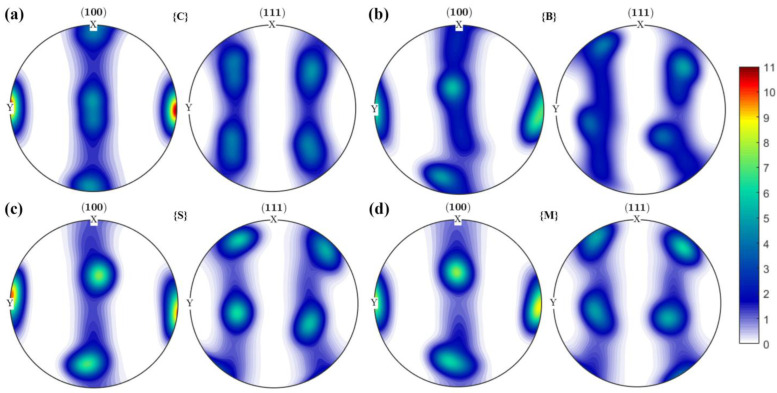
Pole figures of different textures at 10% strain: (**a**) {C}; (**b**) {B}; (**c**) {S}; and (**d**) {M}.

**Figure 5 nanomaterials-13-02214-f005:**
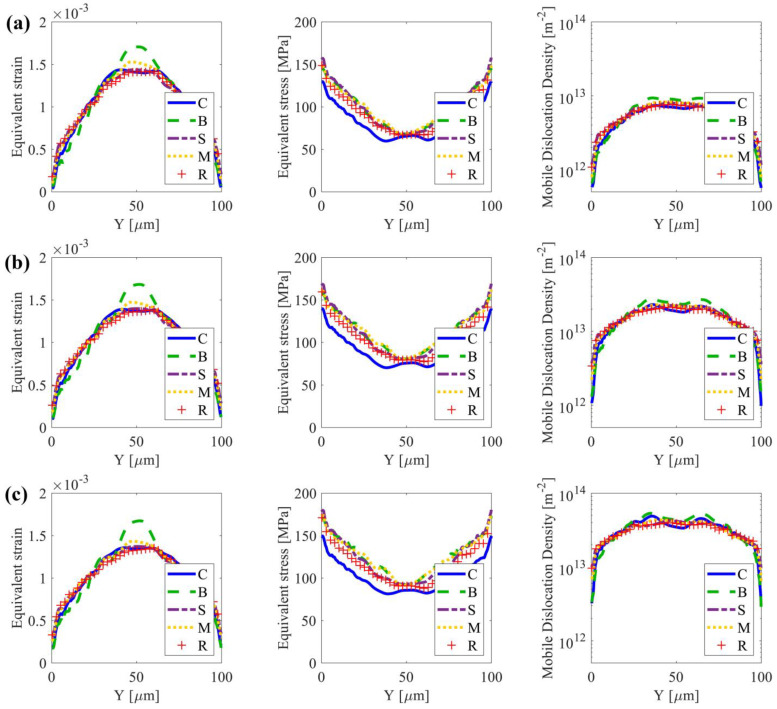
Spatial distributions of equivalent strain, equivalent stress, and mobile dislocation density along the y-direction when D=40 μm. Plots show the average equivalent strain ($\varepsilon_{eq}$), average equivalent stress ($\sigma_{eq}$), and average mobile dislocation density ($\rho_{M}$) at applied strain stages of (**a**) 10%, (**b**) 20%, and (**c**) 30%.

**Figure 6 nanomaterials-13-02214-f006:**
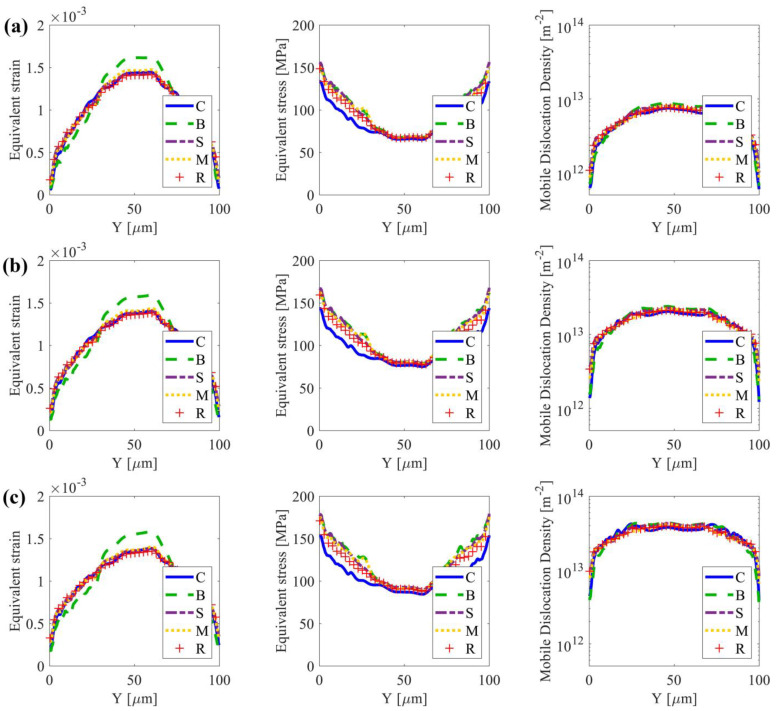
Spatial distributions of equivalent strain, equivalent stress, and mobile dislocation density along the y-direction when D=30 μm. Plots show the average equivalent strain ($\varepsilon_{eq}$), average equivalent stress ($\sigma_{eq}$), and average mobile dislocation density ($\rho_{M}$) at applied strain stages of (**a**) 10%, (**b**) 20%, and (**c**) 30%.

**Table 1 nanomaterials-13-02214-t001:** Common rolling textures for FCC metals [25].

Rolling Texture	{hkl}<uvw>	Euler Angles (°)		
		$\varphi_{1}$	∅	$\varphi_{2}$
C	{001}<100>	0	0	0
B	{011}<211>	0	35	45
S	{123}<634>	64	37	63
M	{112}<111>	90	35	45

**Table 2 nanomaterials-13-02214-t002:** Parameters used in the simulation.

Symbol	Aluminum (Unit)
*c** (Bailey–Hirsch hardening coefficient)	0.35 [45]
$\tau_{0}$ (internal friction)	4.5 MPa [46]
$C_{11}$ (elasticity constant)	108.6 GPa
$C_{12}$ (elasticity constant)	61.3 GPa
$C_{44}$ (elasticity constant)	112 GPa
μ (shear modulus)	28.5 GPa
K (Hall–Petch constant)	0.047 MPa/mm^−1/2^ [46]
$v_{0}$(reference strain rate)	$1\times 10^{-5}$ m/s [45]
1/r (strain rate sensitivity)	0.05 [45]
b (magnitude of Burgers vector)	2.86 Å
$R_{C}$ (critical radius for annihilation coefficient)	15 b [45]
$q_{1},q_{2},q_{3},q_{4},q_{5},q_{6},q_{7}$	0.0325, 2.0, 0.002, 0.002, 0.001, 0.002, 0.1 [35]
$\rho_{M}\rho_{IM}$ (The average mobile, the immobile dislocation density)	$5\times 10^{11}$, $5\times 10^{11}$ $1/m^{2}$
$\Omega^{s\beta }\;$	0.52 (s=β), 0.72 (s≠β) [47]

**Table 3 nanomaterials-13-02214-t003:** Schmid factors of different slip systems for texture {C}, {B}, {S}, and {M} under tension along [010].

Slip System	C	B	S	M
[110](1$\bar{1}$1)	−0.41	−0.27	0	0
[011](1$\bar{1}$1)	−0.41	0	−0.41	−0.41
[10$\bar{1}$](1$\bar{1}$1)	0	−0.27	0.44	0.41
[110](1$\bar{1}\bar{1}$)	−0.41	0.27	0	0
[011](1$\bar{1}\bar{1}$)	−0.41	0	−0.60	−0.41
[101](1$\bar{1}\bar{1}$)	0	0	0.30	0.41
[1$\bar{1}$0](1$1\bar{1}$)	−0.41	0	0.19	0
[011](1$1\bar{1}$)	0.41	0	−0.13	0
[101](1$1\bar{1}$)	0	0	0	0
[1$\bar{1}$0](111)	−0.41	0	−0.10	0
[0$\bar{1}$1](111)	−0.41	−0.28	0	0
[10$\bar{1}$](111)	0	0.28	0	0

**Table 4 nanomaterials-13-02214-t004:** Schmid factors of different slip systems for evolved texture of {C}, {B}, {S}, and {M} at 10% strain.

Slip System	C	B	S	M
[110](1$\bar{1}$1)	−0.40	−0.24	−0.40	−0.41
[011](1$\bar{1}$1)	−0.44	−0.40	−0.46	−0.37
[10$\bar{1}$](1$\bar{1}$1)	0	0.16	0	0
[110](1$\bar{1}\bar{1}$)	−0.41	−0.44	−0.34	−0.39
[011](1$\bar{1}\bar{1}$)	−0.46	−0.73	−0.40	−0.35
[101](1$\bar{1}\bar{1}$)	0	0	0.13	0
[1$\bar{1}$0](1$1\bar{1}$)	−0.39	−0.20	−0.41	−0.41
[011](1$1\bar{1}$)	0.37	0.22	0.31	0.44
[101](1$1\bar{1}$)	0	0	−0.10	0
[1$\bar{1}$0](111)	−0.42	−0.49	−0.33	−0.39
[0$\bar{1}$1](111)	−0.37	−0.27	−0.30	−0.44
[10$\bar{1}$](111)	0	−0.22	0	0

## Data Availability

Data are available on request.

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
