# Peer review of "The Effect of Rolling Texture on the Plastic Deformation of Nano-Gradient Aluminum"

_nanomaterials, 2023, doi:10.3390/nano13152214_

Round 1
Reviewer 1 Report
The paper is aimed at modeling a combined effect of a grain-size gradient and texture gradient on the mechanical properties of aluminum. The results are interesting and the paper can be recommended for publication after correction of the English language and style. The text is written somewhat carelessly. Please, find below some examples. The list is obviously not exhaustive. The text should be reread entirely.
Lines 24-25, “From the "banana curve" [3], it is clear that the 24 gradient materials are balanced due to their strength ductility mutual exclusivity.”
Unclear sentence. Please, reformulate.
Lines 69-71. “Zhang et al. [32] also developed a nonlocal crystal plasticity model to study gradient grained materials with accounting geometrically necessary dislocations and related kinematic hardening, which is able to simulate more number of grains.”
The entire sentence is badly formulated. Besides, “more number of grains” is not a correct sentence. It must be either “more grains” or “a greater number of grains”.
Lines 72-74. “Similarly, grain size effect has also been implemented in the mean field crystal plasticity model [33][34] the visco plastic self-consistent model (VPSC) developed by Lebenson and Tome [35], which can quickly simulate a large number of grains.
Unclear sentence. Perhaps, a conjunction is missed.
Line 101. “…is the can be decomposed as follows.”
‘is the” to be removed.
Line 327. “gradient of aluminum »
The sentence is not rigorous and can be misinterpreted.
Several other lines to correct.
Lines 112,113: “Where ?1-?7 are the coefficient of seven different terms represents different disloca tion interaction mechanisms,…”
Line 157. “by using voronoi tessellation”
Line 336-337. “…the {010} (C) texture can weaken the gradient aluminum on the strength.”
There are also hanging hyphens here and there, e.g., ‘meth-od’, ‘roll-ing’, and so on.
see the text of the review
Reviewer 2 Report
The quality of the manuscript has to be improved.
1) It is necessary to describe in more detail the physical and mathematical formulation of the problem of modeling the tension of model material grain systems using a multilevel approach. It is necessary to indicate the dimension of the problem being solved and the method used for solution of the problem, when using the kinetic version of the constitutive relations.
2) It is necessary to describe in detail the methodology of numerical simulation the response on grain systems to loading. References to articles [34], [35] are not quite correct. In [35], a variant of the model for zirconium alloys with a hexagonal close-packed (HCP) lattice is considered. The physical mechanisms of HCP plastic deformation differ from the deformation mechanisms of aluminum alloys with a face-centered cubic lattice (FCC). Therefore, the validity of the extension of the model to another class of materials belonging to another isomechanical group requires justification. The bibliographic reference [34] has to correct indicating the information about true authors, year, volume and pages of the article. It is necessary to justify the application of the VPSC model presented in the manuscript for FCC aluminum alloys, since the authors of [34] applied the VPSC model developed by Lebensohn and Tome for copper (FCC metal), taking into account the introduced modification Eq.(2) which is the analog of relation (3) in this manuscript. The authors of this manuscript did not use modification Eq. (2) from [34], so the question of the validity of using the VPSC model for aluminum requires additional arguments.
3) The correctness of the presented numerical results should be checked. It is not clear how the authors could obtain the diagrams of engineering stresses versus engineering strains shown in Figures 2 and 3 if these results were obtained by numerical modeling of model grains systems. It is well known that the calculated values of stresses and strains when modeling model samples of materials correspond to the true, but not engineering stresses and strains. The stresses used in equations (2)-(6) correspond to the true, but not engineering ones.
4) It is necessary to indicate the loading conditions of the model grain systems of the material in the calculations, including the strain rate.
5) The using the averaged parameters of the dislocation system for describing the plastic strain of grains in aluminum requires a detailed justification. It is known that during the process of nonlinear strain hardening in aluminum and its alloys, localization bands of plastic deformation at the different scale levels are formed. Localization of plastic deformation is recorded in experiments with quasi-static and high-speed tension of commercially pure aluminum and aluminum alloys.
6) The use of relations (8)-(10) only to take into account the influence of grain boundaries requires additional discussion. It is well known that already at the stage of parabolic strain hardening in FCC metals, dislocation substructures, for example, cellular dislocation substructures, are formed. The boundaries of cellular dislocation structures formed in large grains resist moving dislocations, as well as grain boundaries.
7) It is necessary to indicate all the numerical values of the coefficients and parameters of the model ratios necessary for the reproduction of the calculated results. Table 2 does not list all required coefficients. Reference [38] to the method of fitting the coefficients q1-q7 requires additional justification. In [38], the coefficients were determined using experimental data for Fe-Cr-Ni single crystals. The manuscript deals with polycrystalline aluminum with a gradient grain structure.
8) It is necessary to supplement the manuscript with a detailed presentation of the methodology for the numerical solution of the problem, additionally providing evidence of the numerical convergence of the results obtained. Figure 1 show that in the gradient zone, the sizes of explicitly specified grains change by almost 100 times. Therefore, the spatial discretization step and the corresponding time integration step must be chosen from the condition of correctly describing the deformation of the smallest grains.
9) It is necessary to indicate the ratios for determining the average equivalent strain and average equivalent stress, the values of which are shown in Figures 5 and 6, as well as Figure II and Figure III. Except the stress intensity introduced by Von Mises, other definitions of stress intensity can be used, such as the Hosword equivalent stress. The method for determining the average values of calculated strain and stress parameters was not discussed.
Line 30, line 31: The sentence should be supplemented with missing spaces between words and abbreviations in brackets.
Line 159: The sentence should be supplemented with missing spaces between words.
Line 172. he sentence should be supplemented with missing spaces between words and reference in brackets.
Line 97: The sentence should be supplemented with an indication of the dimension of the reference speed in order to obtain the correct dimension of the Orowan ralation (1).
The grammar corectness of the text of manuscript has to be checked.
Round 2
Reviewer 2 Report
Unfortunately, the authors did not find it possible to correct the obvious errors in the text indicated by the reviewer and supplement the text with the necessary information.
An analysis of the text of the manuscript revised by the authors shows that it quality must be improved.
1) There is no mathematical formulation of the problem using a multilevel approach to modeling the grain tension of a model material. Section 2.1 describes the continuum model of dislocation plasticity VPSC-CDD proposed by Lebensohn and Tome. Model 2D grain structure is specified in subsection 2.3. However, the manuscript does not contain a mathematical formulation of the loading conditions for a model material element, there is no evidence that the considered element is representative, and there is no method for solving the problem and the convergence of numerical values of stresses and strains when varying the parameters of grain structures.
2) It is necessary to describe in detail the methodology of modeling of grain systems to tensile loading. References to articles [34], [35] are not quite correct. In [35], a variant of the model for zirconium alloys with a hexagonal close-packed (HCP) lattice is considered. The physical mechanisms of HCP plastic deformation differ from the deformation mechanisms of aluminum alloys with a face-centered cubic lattice (FCC). Therefore, the validity of the extension of the model to another class of materials belonging to another isomechanical group requires justification. The bibliographic reference [34] has to correct indicating the information about true authors, year, volume and pages of the article. It is necessary to justify the application of the VPSC model presented in the manuscript for FCC aluminum alloys, since the authors of [34] applied the VPSC model developed by Lebensohn and Tome for copper (FCC metal), taking into account the introduced modification Eq.(2) which is the analog of relation (3) in this manuscript. The authors of this manuscript did not use modification Eq. (2) from [34], so the question of the validity of using the VPSC model for aluminum requires additional arguments.
3) The presented numerical results in Figures 2 and 3 should be corrected. It well known the numerical simulation of grains systems loading describes true, but not engineering stresses and strains. The stresses used in equations (2)-(6) correspond to the true, but not engineering ones.
4) The using the averaged parameters of the dislocation system for describing the plastic strain of grains in aluminum requires a detailed justification. It is known that during the process of nonlinear strain hardening in aluminum and its alloys, localization bands of plastic deformation at the different scale levels are formed. Localization of plastic deformation is recorded in experiments with quasi-static and high-speed tension of commercially pure aluminum and aluminum alloys.
5) The use of relations (8)-(10) only to take into account the influence of grain boundaries requires additional discussion. It is well known that already at the stage of parabolic strain hardening in FCC metals, dislocation substructures, for example, cellular dislocation substructures, are formed. The boundaries of cellular dislocation structures formed in large grains resist moving dislocations, as well as grain boundaries.
6) It is necessary to indicate all the numerical values of the coefficients and parameters of the model ratios necessary for the reproduction of the calculated results. Table 2 does not list all required coefficients. Reference [38] to the method of fitting the coefficients q1-q7 requires additional justification. In [38], the coefficients were determined using experimental data for FeCr-Ni single crystals. The manuscript deals with polycrystalline aluminum with a gradient grain structure.
7) It is necessary to indicate the ratios for determining the average equivalent strain and average equivalent stress, the values of which are shown in Figures 5 and 6, as well as Figure II and Figure III. Except the stress intensity introduced by Von Mises, other definitions of stress intensity can be used, such as the Hosword equivalent stress. The method for determining the average values of calculated strain and stress parameters was not discussed.

Author Response
Please see the attached. We highlight our revision in yellow.

Round 3
Reviewer 2 Report
The authors carried out a significant revision of the text of the manuscript, as a result of which the quality of the manuscript improved significantly. Unfortunately, some of the reviewer's comments were not correctly understood and the text of the manuscript contains incorrect provisions.
In this regard, the manuscript has to be improved.
1) The engineering stresses versus engineering strains are shown in Figures 2 and 3. The authors added the explanation to the manuscript that engineering stresses versus engineering strains were recalculated from calculated true stresses and strains (Line 208). However, the application of the relations under the considered deformation conditions is not correct. This is due to the fact that the used formulas relating the true and engineering stresses and strains can be applied in a limited the range of strain. This range includes only a uniform distribution of strains in the gage zone of the samples or simulated zone of alloy with heterogeneous grain structure. The presented diagrams of engineering stresses and strains at strains above 0.18 ... 0.2 have descending branches, which is associated with the inhomogeneity of the strain field in the model computational textured zone of alloy. Thus, the data presented in Figures 2 and 3 are not correct for strains exceeding 0.18…0.2. If the authors had presented the calculated values of the true stresses and strains, then the error could have been avoided.
2) The spatial distributions of equivalent strain and equivalent stress are shown in Figures 5 and 6, as well as Figure II and Figure III. But authors did not show the relations for calculation of equivalent strain and equivalent stress. It well known that except the stress intensity introduced by Von Mises, the Hosword equivalent stress is uses too.
The Hosword equivalent stress is often used in the yield criteria for textured and anisotropic alloys. Therefore, the indication of specific formulas for calculating equivalent strain and equivalent stress is necessary in the manuscript.
